# Vascular Proximity Increases the Risk of Local Recurrence in Soft-Tissue Sarcomas of the Thigh—A Retrospective MRI Study

**DOI:** 10.3390/cancers13246325

**Published:** 2021-12-16

**Authors:** Andrea Sambri, Emilia Caldari, Andrea Montanari, Michele Fiore, Luca Cevolani, Federico Ponti, Valerio D’Agostino, Giuseppe Bianchi, Marco Miceli, Paolo Spinnato, Massimiliano De Paolis, Davide Maria Donati

**Affiliations:** 1Department of Orthopaedics, Istituto di Ricerca e Cura a Carattere Scientifico (IRCCS), Azienda Ospedaliera, Universitaria di Bologna, 40138 Bologna, Italy; emiliacaldari@gmail.com (E.C.); massimiliano.depaolis@aosp.bo.it (M.D.P.); 2Dipartimento di Scienze Biomediche e Neuromotorie (DIBINEM), University of Bologna, 40126 Bologna, Italy; andrea.montanari36@studio.unibo.it (A.M.); michele.fiore@ior.it (M.F.); davide.donati@ior.it (D.M.D.); 3Istituto Ortopedico Rizzoli, Istituto di Ricerca e Cura a Carattere Scientifico (IRCCS), 40136 Bologna, Italy; luca.cevolani@ior.it (L.C.); federico.ponti@ior.it (F.P.); valeriodrago@libero.it (V.D.); giuseppe.bianchi@ior.it (G.B.); marco.miceli@ior.it (M.M.); paolo.spinnato@ior.it (P.S.)

**Keywords:** soft tissue sarcoma, recurrence, vascular proximity, by-pass, MRI

## Abstract

**Simple Summary:**

Proximity to major vessels increases risk of local recurrence in soft tissue sarcomas of the thigh. When major vessels were observed to be surrounded by the tumor on preoperative MRI, vascular resection and by-pass reconstruction offered a better local control.

**Abstract:**

The aim of this study was to establish the prognostic effects of the proximity of the tumor to the main vessels in patients affected by soft tissue sarcomas (STS) of the thigh. A total of 529 adult patients with deeply seated STS of the thigh and popliteal fossa were included. Vascular proximity was defined on MRI: type 1 > 5 mm; type 2 ≤ 5 mm and >0 mm; type 3 close to the tumor; type 4 enclosed by the tumor. Proximity to major vessels type 1–2 had a local recurrence (LR) rate lower than type 3–4 (*p* < 0.001). In type 4, vascular by-pass reduced LR risk. On multivariate analysis infiltrative histotypes, high FNLCC grade, radiotherapy administration, and type 3–4 of proximity to major vessels were found to be independent prognostic factors for LR. We observed an augmented risk of recurrence, but not of survival as the tumor was near to the major vessels. When major vessels were found to be surrounded by the tumor on preoperative MRI, vascular resection and bypass reconstruction offered a better local control.

## 1. Introduction

Soft tissue sarcomas (STS) of the limbs include a large variety of histologies, sizes, and grades, with clear and significant variances in their clinical course and prognosis [1]. Despite these intrinsic varieties, STS are very rare tumors, which represent only 1% of all malignancies considering all locations in the adult population. These variabilities, together with a lack of knowledge of these rare cancers among general practitioners, often lead to a delayed presentation at referral centers [2]. A delayed diagnosis can cause the observation of large lumps at baseline, thus increasing difficulties in surgical treatment. Therefore, an early diagnosis and a multidisciplinary approach are mandatory to choose the best strategy for each patient.

Limb salvage surgery with radiation therapy (RT) is the standard of care for most patients, while the role of chemotherapy (ChT) is more debated [3]. There is still disagreement on which are the most appropriate surgical strategies when the tumor involves the principal vessels of the limb.

A decision must be made between an extended resection with the risk of causing a deficiency for the patient or limiting the excision with the risk of incomplete tumor removal [4].

Magnetic Resonance Imaging (MRI) is the imaging tool of choice to evaluate STS of the extremities, with fundamental information in all the phases of the disease, in particular for surgical planning [5]. An accurate MRI study at baseline is imperative for decision-making. Several MRI features can change drastically the surgical approach, above all the identification of the so-called “tail sign”, which can be observed in some “infiltrative” histotypes [6]. In these cases, a wider resection is suggested [7]. In addition, MRI is the imaging of choice to assess the relationship between the tumor mass and vascular bundles.

To date, many clinical studies tried to identify prognostic factors for local recurrence (LR) and overall survival (OS). Numerous factors have been suggested as possible risk factors for developing LR or distant metastasis, including histological STS-subtypes, surgical margins, tumor size, grade, administration of neo(adjuvant) RT or ChT [8,9,10,11,12,13]. However, an established prognostic profile for individual patients is still lacking, even though the risk factors for the occurrence of LR and distant metastasis have been studied in many series [14].

Nonetheless, the correlation between the closeness between major vessels and the tumor and outcomes (LR and survival) has never been investigated in STS. Recently, Fujiwara et al. [15] observed that the proximity to major blood vessels is a poor prognostic factor for local control and survival in patients affected by osteosarcoma.

The aim of this study was to explore the influence of the proximity of the tumor to the main vascular structures in a large population with deep STS of the thigh and popliteal fossa.

## 2. Material and Methods

### Cases

All patients with a diagnosis of STS treated at a single Institution who underwent surgery from January 2000 to April 2017 were retrospectively analyzed.

Inclusion criteria included primary, deeply seated STS, location in the thigh and popliteal fossa, age ≥18 years and pre-operative MRI available for retrospective evaluation. Depths were evaluated according to preoperative MRI. Patients were excluded if treated with an amputation to analyze a more homogeneous series. Thus, a total of 529 patients meeting these criteria were selected from the whole series.

The study was approved by the local Ethics Committee.

All cases were histologically revised and classified according to the 2013 World Health Organization classification of STS [16]. A 3-step system (FNCLCC) was used to assess STS grade [16].

Some STS subtypes (myxofibrosarcoma and undifferentiated pleomorphic sarcoma) may present a specific focal-infiltrative pattern and were therefore classified as “infiltrative subtypes” [7].

The larger diameter was measured on the surgical specimen to assess tumor size. All patients underwent surgery in order to obtain limb-sparing, function-sparing surgery with negative surgical margins according to the R classification [17].

The use of RT and ChT was decided at the discretion of a multidisciplinary team (orthopedic surgeon, radiotherapist, and oncologist). Radiation therapy and ChT were administrated according to STS guidelines [18].

All patients had an MRI performed within a maximum of 21 days prior to surgery. All MRI were performed with a high field (1.5 Tesla or higher) and with the standard sequences, which included T1w, T2w, STIR, or T2 fat sat. Magnetic resonance imagines were retrospectively reviewed on a PACS (Carestream Vue PACS v. 11.4.1.1102) by two expert oncologic radiologists (PS and FP). Imaging analyses were performed in consensus. The closeness to major vessels was defined as the minimum distance between the tumor and the main vascular bundle. This distance was measured on the axial slices of preoperative T1-weighted MRI [19]. Distance between the major vascular bundle and the tumor was classified into four types according to Fujiwara et al. [15]: type 1, >5 mm; type 2, ≤5 mm and >0 mm; type 3, attached to the tumor; type 4, surrounded by the tumor (Figure 1).

Patients’ characteristics are presented by frequencies and percentages for categorical variables, median, and range for continuous variables.

The Kaplan–Meier method was used to estimate OS and LR-free survival. The Kaplan–Meier method was used as a univariate unadjusted analysis of prognostic factors to estimate LR-free survival and OS. The estimated local recurrence-free survival interval was defined as the time between surgery and first LR. Being a retrospective study, patients were censored at the last available follow-up, in case of death, or in case of amputation not related to local recurrence. Overall survival interval was defined as the time between surgery and death, during the time of follow-up. Patients who died of other causes were censored. Differences in survival rates were assessed by the log-rank test. Multivariable analysis of LR and OS was based on cause-specific hazards and therefore carried out by Cox regression models. The following factors were included in multivariate analysis: histotype, FNLCC grade, proximity to major vessels, radiotherapy, chemotherapy, tumour size, age, and margins. *p* values < 0.05 were considered significant. All analysis was completed using the Statistical Package for Social Science (IBM Corp. Released 2013. IBM SPSS Statistics for Windows, Version 22.0. Armonk, NY, USA).

## 3. Results

Median age at the time of surgical procedures was 58 years (range, 18 to 92). (Table 1) Most of the patients (288, 54.4%) were male and 241 (45.6%) were female.

The majority of STS were grade 3 (421, 79.6%), 9.6% were grade 2, and 10.8% grade 1. Size of the tumour was larger than 10 cm in 47.4% of the cases, 5–10 cm in 36.5%, and smaller than 5 cm in 16.1%. Non-infiltrative histotypes accounted for 66.5% of the whole series, whereas in 33.5% were infiltrative STS.

Type 1 involvement was found in 285 (53.9%) of the cases on pre-operative MRI, whereas 61 (11.5%) were type 2. The tumour was attached to major vessels (type 3) in 108 (20.4%) of the cases and surrounded by the tumor 75 (14.2%). Type 3–4 vessels involvement were mostly observed among non-infiltrative STS (*p* = 0.030) and large tumors (*p* < 0.001). Fifty-eight patients (11.0%) were metastatic at the time of diagnosis (43 to the lungs, in 7 cases to lymph nodes, 2 cases of bone metastasis, and 5 to the liver). A vascular by-pass was required in 21 (4.0%) of the cases, mostly in type 3–4 vessels involvement (*p* < 0.001). All the cases had only artery reconstructed.

Type 3–4 vessels involvement was mostly found in STS larger than 10 cm (*p* < 0.001) and in non-infiltrative histotypes (*p* = 0.030). However, no correlation was found with FNLCC grade (*p* = 0.258).

R0 margins were obtained in 494 cases (93.3%), R1 margins in 35 (6.7%). No correlation was found between the quality of surgical margins and type of vessels involvement (*p* = 0.288). No amelioration in the quality of surgical margin was observed in patients who underwent artery sacrifice and reconstruction with a by-pass (*p* = 0.727).

Chemotherapy was used in 253 (47.8%) STS, mostly in non-infiltrative histotypes (*p* = 0.021), in patients with metastasis at presentation (*p* < 0.001), in grade 3 tumors (*p* < 0.001), and in those presenting with a type 3–4 vessels involvement (*p* = 0.005). Patients younger than 65 years at the time of diagnosis received ChT more frequently (*p* < 0.001).

In 369 (69.8%) cases, RT was given in addition to surgery as part of the primary treatment, mainly in grade 3 STS (*p* < 0.001), large tumors (*p* < 0.001), and in those with a pre-operative type 3–4 vessels involvement (*p* = 0.003). Radiotherapy was administered in the pre-operative setting in 94 cases, with no amelioration of surgical margins adequacy (*p* = 0.645).

At the last follow-up (median 31 months, range 2 to 269), 363 patients (68.6%) were alive; 107 (20.2%) died of the disease; 10 patients (1.9%) died of other causes. Forty-nine patients (9.3%) were alive with progressive metastatic disease.

### 3.1. Local Recurrence

In 86 cases (16.3%), a LR occurred after a median time of 17 months (range, 2 to 96).

Local recurrence-rate was 19.1% (C.I. 95% 15.0%–23.2%) at five years and 41.4% (C.I. 95% 23.8–39.0%) at ten years.

A higher LR rate was found in infiltrative STS subtype (29.6% vs. 14.5% at five years, *p* = 0.005). (Figure 2) FNLCC grade 2 and 3 STS had also an increased risk of LR (15.5% and 21.8% at five years, respectively) compared to grade 1 (1.8% at 5 years) (*p* = 0.035). Proximity to major vessels type 1 and 2 had a LR rate lower (12.0% and 12.4% at five years, respectively) than type 3 and 4 (26.3% and 40.6% at five years, respectively) (*p* < 0.001). No significant differences were observed neither for size of the tumor (*p* = 0.156) nor for RT administration (*p* = 0.181) with LR rate. Patients with adequate surgical margins had a lower LR-rate, even though this did not reach statistical significance (*p* = 0.069). No amelioration of local control of the disease was observed among patients treated with a vascular by-pass (*p* = 0.223). However, among type 4 vascular involvement cases, a vascular by-pass allowed for an amelioration in local control of the disease (LR rate 21.2% at five years, lower than in type 3 vascular involvement without vascular sacrifice).

On multivariate analysis infiltrative histotypes, high FNLCC grade and type 3–4 of proximity to major vessels were confirmed to be independent prognostic factors for LR (Table 2). Radiotherapy was also found to be an independent prognostic factor for LR.

### 3.2. Sarcoma Specific Survival

Estimated overall survival, measured with a Kaplan–Meier analysis was 73.1% (C.I. 95% 68.2–78.0%) at five years and 63.2% (C.I. 95% 56.4–70.0%) at ten years.

A significant worse OS was observed in STS with distant metastasis at presentation (20.4% vs. 79.5% at five years follow up, *p* < 0.001). Patients treated with ChT had a better OS (66.6% vs. 80.3% at five years, *p* = 0.001) (Figure 3).

Grade 2 and 3 STS had an inferior prognosis (OS 78.6% and 69.1% at five years, respectively) if compared to grade 1 (100% OS at five years, *p* < 0.001). Regarding histology, the worst OS was found in infiltrative over non-infiltrative subtypes (65.6% vs. 76.7 at five years, *p* = 0.005). Patients with a major vessels involvement type 1 and 2 had a better OS (77.5%, 81.7%, respectively, at five years) compared to type 3 and 4 (69.5%, 55.9, respectively, at five years) (*p* = 0.008).

On multivariate analysis, proximity to major vessels was not confirmed to be an independent prognostic factor for OS (Table 3).

## 4. Discussion

One of the most important factors to be considered by surgeons when treating patients affected by STS is the distance between the tumor and the major vessels. However, to the best of our knowledge, no previous study focused specifically on the prognostic implication of the proximity of the tumor to major vessels in STS.

In the present series, proximity to major vessels was observed to be an independent prognostic factor for local recurrence in primary STS of the thigh and popliteal fossa, thus providing a possible prognostic estimation based on preoperative MRI assessment.

Moreover, we confirm that two of the most important predictors of outcome are histology/histologic subtype and grade [1]. Specific infiltrative STS subtypes such as myxofibrosarcoma and undifferentiated pleomorphic sarcoma have an increased risk of LR regardless of the adequacy of surgical margins [7,20,21,22]. A non-significant difference in the LR rate was observed concerning the adequacy of surgical margins. However, it must be considered that the interpretation of the surgical margins can be subjective and might vary depending on who assessed the specimen.

Radiation therapy seemed to have a strong impact to reduce the risk of a recurrence. Moreover, patients who received RT were unfavorably selected, the choice being a clinical one performed on an individual basis. For example, RTE was mostly administered in large and type 3–4 tumors.

Overall survival was related to the histological tumor grade. The main difference in outcome was identified between grade 2 and grade 3 tumors, whereas less important differences were observed between grade 1 and 2 STS. All the patients affected by grade 1 sarcoma were alive at the final follow-up. Chemotherapy was not found to ameliorate OS on multivariate analysis. However, patients who underwent ChT were unfavorably selected, being selected for ChT those at a higher risk. In particular, patients older than 65 years received ChT in fewer cases than younger patients. Proximity to major vessels was not found to influence OS on multivariate analysis. Biological aggressiveness of the tumor rather than vessel proximity is the major factor behind patient survival. The age of the patient was also found to be an independent prognostic factor for survival. Acem et al. previously observed that this can only partially be explained by differences in tumor and treatment characteristics, suggesting that STS may have a more aggressive tumor behavior in elderly patients when compared with their younger counterparts, which may coincide with a weaker tumor-specific immune response in elderly patients [11].

Most of the deeply seated STS of the thigh and popliteal fossa included in the present series had a distance to major vessels >5 mm. Nonetheless, approximately one third of the cases were attached or surrounded vascular bundle (type 3 and 4, respectively), particularly evident among large tumors and non-infiltrative histotypes. In many cases, the main vessels are just pushed by the STS, but not actually penetrated. Thus, surgery with additional complex resection of the vessels may represent an overtreatment in these situations, growing the risk for an unnecessarily increased risk of surgical morbidity. Nonetheless, when a major vascular bundle is found to surround the sarcoma on preoperative MRI, a vascular by-pass reduced local recurrence risk, with no amelioration on survival rate. A combined ortho-vascular approach for STS patients is a challenge and can increase the incidence of postoperative complications, but can help in ameliorating surgical outcomes. [4,23,24]

In our opinion, these results may offer valuable prognostic information for treating oncologists and could be helpful to advise patients on the management of their STS. The strategy of treatment should be tailored for each case, considering the possible survival benefits, and expected morbidity.

The retrospective design of this study is one of the main limitations, with possible selection biases regarding diagnosis, treatment, and follow-up of patients included.

Moreover, margins classification on a retrospective series could have been affected by their previous assessment. In addition, the artery and vein were considered together as “major vessels”.

However, this series includes data from a selected cohort of patients, which include only adults affected by primary sarcomas of the thigh and popliteal fossa. A limited follow-up in some of the patients could also slightly affect the results. Moreover, the quality of the MRI was not always consistent between patients. We believe that a prospective study with matching between the magnetic resonance images, the pathological reports and excised gross specimen, as well as a precise lesion-by-lesion correlation would be necessary. Recent advantages in regards of quantitative imaging may offer more precise and reliable information, in all the phases of the disease. Several quantitative tools applicable to MRI such as diffusion (DWI) sequences, dynamic contrast-enhanced, and radiomics analyses are increasing in use to obtain a solid and more reliable evaluation [25,26]. In the near future, the radiomic features extrapolated from MRI studies will permit to build artificial intelligence algorithms, in order to help in diagnosis, to predict patients’ prognosis and for tumor grade prediction, and to evaluate responses to treatments.

In the literature, there is a common lack of awareness about the importance of the tumor-vascular relationship as a prognostic factor for local recurrence. The lack of association between tumor-vascular encasement and overall sarcoma-specific survival, confirmed the results similarly obtained by Crombé et al [26]. in a smaller sample of patients [27]. On the contrary, to our knowledge, the association observed in the present series between tumor-vascular proximity and the increased risk of LR has never been reported before in patients affected by STS.

Our results encourage us to consider in the conventional MRI analysis tumor proximity to major vessels. Tumor-vascular closeness must be evaluated, quantified, and considered among the other already known MRI features related to patients’ prognosis, which include peritumoral enhancement, signs of necrosis, signal intensity inhomogeneity, ill-defined borders, and signs of infiltrations [27,28,29,30].

In the case of patients with contraindications to MRI exams, the same evaluation in regards to tumor-vascular proximity can be reached with contrast-enhanced computed tomography.

## 5. Conclusions

In conclusion, our experience seems to confirm that the closeness between STS and major blood vessels can predict a poorer local control, but not survival.

We observed an increased risk of local recurrence as the tumor developed adjacent to the major vessels, despite the use of radiation therapy. Even if RTE can independently reduce the risk of LR, in the case of pre-operative MRI findings of the tumor attached to the major vascular bundle, a vascular resection and by-pass reconstruction offered a better local control.

Stratification of risks is mandatory and can contribute to an amelioration of the treatment decisions, thus improving clinical results in patients affected by STS.

## Figures and Tables

**Figure 1 cancers-13-06325-f001:**
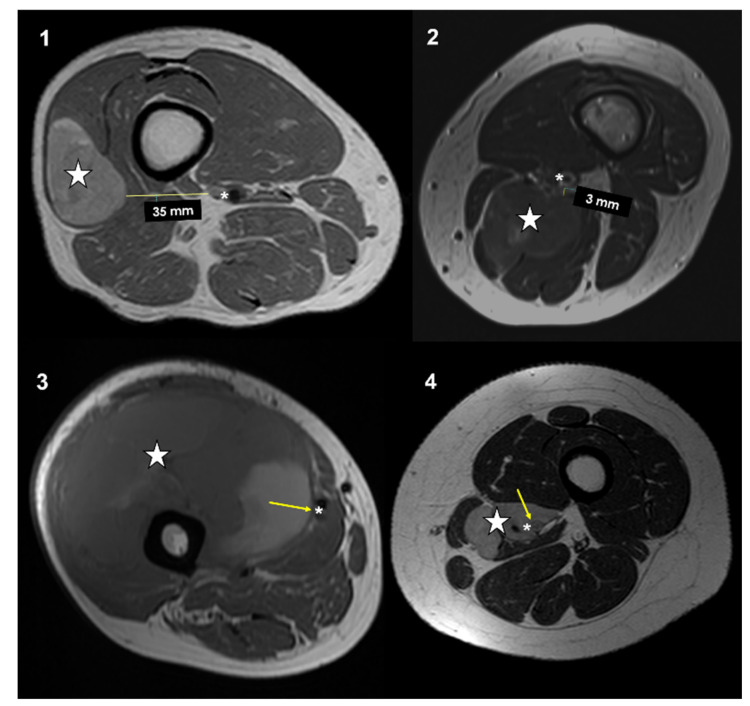
Types of vascular proximity based on MRI T1w axial images performed in four different patients affected by soft-tissue sarcomas of the thigh and popliteal fossa: type 1: >5 mm; type 2: >0 mm and <5 mm; type 3: in contact; type 4: surrounded. Stars (neoplasms), asterisks (main vessels), arrows (tumor-vascular contact).

**Figure 2 cancers-13-06325-f002:**
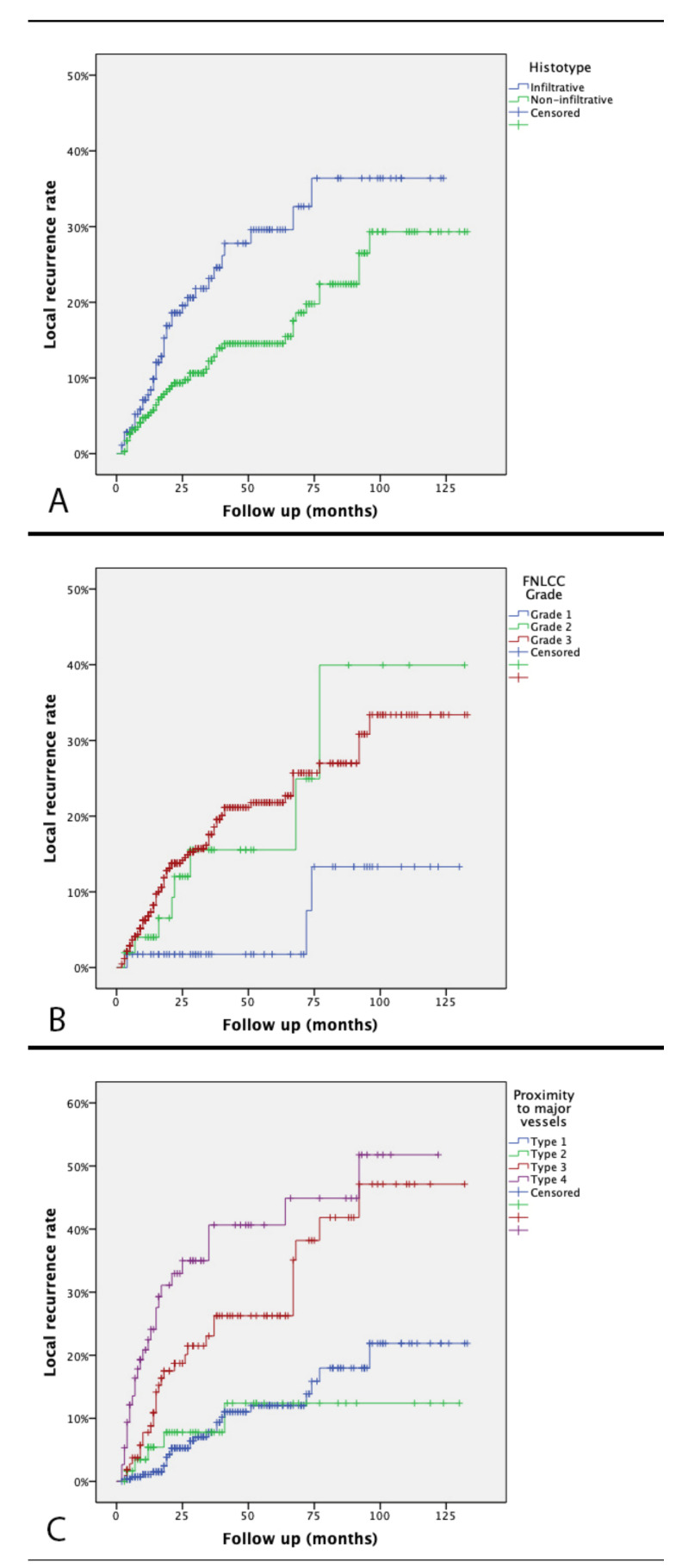
Kaplan–Meier local recurrence curve according to histotype (**A**), FNLCC grade (**B**), and proximity to major vessels (**C**).

**Figure 3 cancers-13-06325-f003:**
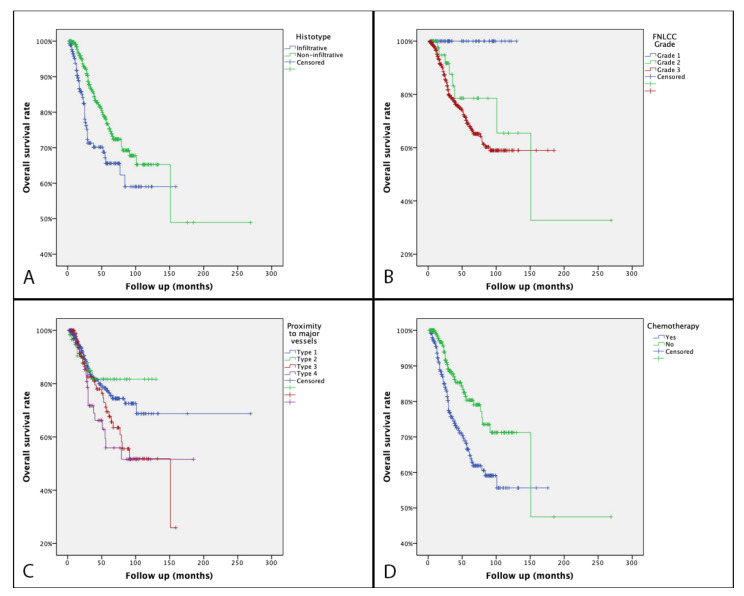
Kaplan–Meier overall survival curve according to histotype (**A**), FNLCC grade (**B**), proximity to major vessels (**C**), and chemotherapy (**D**).

**Table 1 cancers-13-06325-t001:** Patients’ characteristics.

Characteristics	*N* (%)
Age	Median 58 years (range, 18–92)
Sex	
Male	288 (54.4%)
Female	241 (45.6%)
FNLCC Grade	
Grade 1	57 (10.8%)
Grade 2	51 (9.6%)
Grade 3	421 (79.6%)
Size	
<5 cm	85 (16.1%)
>5 cm	193 (36.5%)
>10 cm	251 (47.4%)
Histotype	
Infiltrative	177 (33.5%)
Non-infiltrative	352 (66.5%)
Vessels involvement	
Type 1	285 (53.9%)
Type 2	61 (11.5%)
Type 3	108 (20.4%)
Type 4	75 (14.2%)

**Table 2 cancers-13-06325-t002:** Multivariate analysis for local recurrence.

Characteristics	HR	95% CI	*p* ^a^
Histotype:	1.998	1.294–3.098	0.002
(Infiltrative vs. Non-infiltrative)			
FNLCC Grade	1.772	1.134–2.769	0.012
(2–3 vs. 1)			
Proximity to major Vessels	0.198	0.124–0.317	<0.001
(type 1–2 vs. type 3–4)			
Radiotherapy	0.472	0.298–0.750	0.001
(Yes vs. No)			
Chemotherapy	0.767	0.480–1.223	0.265
(Yes vs. No)			
Margins	0.712	0.360–1.408	0.329
(R0 vs. R1–R2)			
Age	1.116	0.653–1.906	0.688
(≤65 vs. >65 years)			
Size	1.051	0.680–1.625	0.822
(≤10 vs. >10 cm)			

HR hazard ratio, 95% CI 95% confidence interval. ^a^ backward Wald test.

**Table 3 cancers-13-06325-t003:** Multivariate analysis for overall survival.

Characteristics	HR	95% CI	*p* ^a^
Histotype	1.520	1.014–2.281	0.043
(Infiltrative vs. Non-infiltrative)			
Distant metastasis at presentation	2.670	2.136–3.298	<0.001
(Yes vs. No)			
FNLCC Grade	2.205	1.206–4.031	0.010
(2–3 vs. 1)			
Proximity to major Vessels	0.780	0.643–1.098	0.054
(type 1–2 vs. type 3–4)			
Chemotherapy	1.523	0.992–2.316	0.059
(Yes vs. No)			
Radiotherapy	0.852	0.533–1.364	0.506
(Yes vs. No)			
Margins	0.688	0.355–1.334	0.269
(R0 vs. R1–R2)			
Age	0.448	0.285–0.704	0.001
(≤65 vs. >65 years)			
Size	0.897	0.609–1.322	0.583
(≤10 vs. >10 cm)			

HR hazard ratio, 95% CI 95% confidence interval. ^a^ backward Wald test.

## Data Availability

The data presented in this study are available on request from the corresponding author.

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
