# Peer review of "Vascular Proximity Increases the Risk of Local Recurrence in Soft-Tissue Sarcomas of the Thigh—A Retrospective MRI Study"

_cancers, 2021, doi:10.3390/cancers13246325_

Round 1

Reviewer 1 Report

Many thinks for this very well written and informative study on the relation of vessels to STS and the influence on outcome.

I have a couple of suggestions to make the paper statistically more sound and prevent for conclusions being based on the classic difficulties of retrospective analyses. please consider to adhere to some of them before re-submitting the paper to Cancers

Please use a competing risk analyses for LR as one patient cannot have a LR when dead of disease and 40% is dead of disease after 5 years

Please use R0-1-2 margin descriptions as they are probably more informative than Ennekings and more predictive for LR and OS

Please inform the reader on the multivariate analyses in what parameters are in the analyses as now it seems as if RxTh and margins are not in the Multivariate analyses.

Was the use of chemotherapy a predictive value due to inclusion bias ? it seems that healthy and young patients would indeed get Chemo and the elderly that are more likely to die within 5 years are not ?

 Please elaborate on the cross correlation between size and type and relation to the vessels ?

One could also argue that high grade, infiltrative large tumors have an intrinsic higher chance of being close to the vessels and therefor need either resection of the vessels or (neo)adjuvant RxTh/Chemo to improve the outcome both on Local control and OS? And maybe its not the relation between vessels and STS but the biological intrisic risk of the STS that is displayed on MRI by being close to the vessels.

I would love to see a clear analyses of the influence of RxTh and vessel reconstruction on the OS en LR rates in order to present a well balanced advise for the readers on when to resect and when just to add Rxth to get better Local control when the tumor is close to vessels. Also close to the vein is different from close to artery ??

Hope to read the paper again after these comments are possibly incorporated in the paper.

Author Response

  • Please use competing risk analyses for LR as one patient cannot have a LR when dead of disease and 40% is dead of disease after 5 years

Thank you very much for the interesting observation, which allows us to formulate our opinion on the interpretation of these data. In standard survival analysis, the event-free time of subjects who do not experience the outcome of interest during the observation period is generally censored at the end of follow-up. However, censoring may occur for various reasons. A patient may be lost to follow-up during the study or may experience another event that makes further follow-up impossible or unnecessary. An important assumption of standard survival analysis methods such as the Kaplan-Meier method is that censoring is "independent" [Lau B et al. Competing risk regression models for epidemiologic data, Am J Epidemiol, 2009] of the event of interest. Based on this assumption, we censored patients who died during follow-up, patients who were lost to follow-up, and patients who underwent amputations for reasons unrelated to recurrence, considering all these reasons to be unrelated to local recurrence. In fact, in our opinion, local recurrence may be associated with overall survival as a predictive factor, not as an independent agent of death. Therefore, we believe that a competing cumulative incidence risk method is not necessary to provide a reliable estimate of the risk of recurrence in the context of univariate unadjusted analyses of prognostic factors. Nevertheless, we agree that the description of the analysis of the estimated local recurrence risk in the methods section appears to be inaccurate in its current form.

Therefore, we modified it within the methods section.

  • Please use R0-1-2 margin descriptions as they are probably more informative than Ennekings and more predictive for LR and OS

Margins were re-classified using the Residual tumour classification, according to this comment.

  • Please inform the reader on the multivariate analyses in what parameters are in the analyses as now it seems as if RxTh and margins are not in the Multivariate analyses.

Thank you for this comment. The methods section has been modified accordingly.

  • Was the use of chemotherapy a predictive value due to inclusion bias ? it seems that healthy and young patients would indeed get Chemo and the elderly that are more likely to die within 5 years are not ?

Chemotherapy was used mostly in non-infiltrative histotypes, in patients with metastasis at presentation, in grade 3 tumors, and in those presenting with a type 3-4 vessels involvement.

These selection biases may explain why patients who underwent ChT had worse survival. This point has been discussed within the manuscript.

  • Please elaborate on the cross correlation between size and type and relation to the vessels?

Thank you for this comment. This part was added to the results.

  • One could also argue that high-grade, infiltrative large tumors have an intrinsic higher chance of being close to the vessels and therefore need either resection of the vessels or (neo)adjuvant RxTh/Chemo to improve the outcome both on Local control and OS? And maybe it’s not the relation between vessels and STS but the biological intrinsic risk of the STS that is displayed on MRI by being close to the vessels tumour, which is the major factor behind patient survival. 

Thank you for this comment. Large and infiltrative STS were found to be associated with closeness to the major vessels. However, no correlation was found regarding FNLCC grade. As highlighted by multivariate analysis, we observed that proximity to major vessels can predict the risk of LR but not OS. On the other hand, biological aggressiveness rather than closeness to vessels can predict survival.

This part has been discussed within the manuscript.

  • I would love to see a clear analysis of the influence of RxTh and vessel reconstruction on the OS and LR rates in order to present well-balanced advice for the readers on when to resect and when just to add Rxth to get better Local control when the tumor is close to vessels.

Thank you for this comment which has led us to a deeper reflection on the statistical analysis conducted. Initially, we did not include radiotherapy in the analysis because of its non-significance in the univariable analysis [Karcutskie et al. JAMA Surg. 2017; Templin et al. N Engl J Med. 2015; Nor et al. Lancet Neurol. 2005]. However, we acknowledge that this is a simplified method widely used in biomedical research published in top medical journals to select the variables included in the multivariable analysis. We agree that this method has considerable limitations and exposes to the real risk of missing important elements as there are variables that may not be important in a univariable association but are important in the full model. [Sun et al. J Clin Epidemiol 1996; Harrel et al. Regression Modeling Strategies: With Applications to Linear Models, Logistic Regression and Survival Analysis. New York, NY: Springer-Verlag; 2001; Heinze et al. Transpl Int 2017]

In fact, there are lots of reasons to include non-significant variables as they mediate and moderate other relationships, theory says that they ought to be in the model, the effect size is large, even if it is not significant, it was part of your a priori hypothesis, etc. However, the process of selecting variables is very arduous and there is no agreement on the strategy to be used, as the most appropriate way can vary greatly according to different problems and disciplines. Commonly used methods for Cox regression model are forward selection, backward elimination, stepwise selection, purposeful selection algorithms, best subset selections and least absolute shrinkage and selection operator. [Bursac et al. Source Code Biol Med. 2008; Garcia et al. Biometrics. 2010].

Some methodologists suggest including all clinical and other relevant variables in the model regardless of their significance in order to control for confounding. This approach, however, can lead to numerically unstable estimates and large standard errors. Nevertheless, for this article, after statistical review, as suggested, we decided to include in the multivariable analysis all variables analyzed in the univariable, regardless of their significance, as they are all potentially clinically relevant. As a result, we found radiotherapy as a significant variable on multivariable analysis, although not significant on univariable. This is not an uncommon occurrence in this type of analysis, as highlighted in the literature [Lo et al. Changgeng Yi Xue Za Zhi. 1995; Wang et al. Shanghai Arch Psychiatry. 2017] and can be seen as evidence of the power of multivariable analysis in detecting confounding factors.

We would like to conclude by thanking you again for this comment which has led to a considerable improvement in the quality of the evidence for this study.

The manuscript has been modified accordingly.

  • Also close to the vein is different from close to artery ??

Thank you again for the precious comment. In this retrospective analysis, artery and vein involvement were considered together as “major vessels involvement”. In the case of vascular bypass, all cases had only artery reconstructed. Thus, we believe that the identification of artery involvement by STS is more important.

Reviewer 2 Report

This is a retrospective review of MRI imaging of soft-tissue sarcomas of the thigh and knee and correlation of the proximity of the vessels to the local recurrence rate. The previous literature supports that surgical margins are important in the local control of the disease, although major discrepancies regarding surgical margin classification render generalisation of findings difficult.  Radiological and surgical margins are obviously correlated and confounders. 

In this study, which includes a relatively large number of patients, surgical margins were found not to have a statistically significant effect on recurrence rate, and thus excluded from multivariable analysis. This observation may be due to the choice of subgroups, i.e. marginal resection been classified as non-adequate. In any case, multivariable analysis should include margins (all 3 groups included) and radiotherapy. 

Minor points: 

-In the discussion, the authors postulate that radiological classification did not have an effect on overall survival due to the protective effect of chemotherapy. This is far-reached, the simple explanation is that vessel proximity has nothing to do with the biological aggressiveness of the tumour, which is the major factor behind patient survival. 

Author Response

  • In this study, which includes a relatively large number of patients, surgical margins were found not to have a statistically significant effect on recurrence rate, and thus excluded from multivariable analysis. This observation may be due to the choice of subgroups, i.e. marginal resection been classified as non-adequate. In any case, multivariable analysis should include margins (all 3 groups included) and radiotherapy. 

We thank the reviewer for the precious comment. Margins were re-classified according to the Residual tumour classification. However, we acknowledge that margins assessment in retrospective series can be influenced by major bias. This point has been discussed as a limitation.

Thank you for this comment which has led us to a deeper reflection on the statistical analysis conducted. Initially, we did not include radiotherapy in the analysis because of its non-significance in the univariable analysis [Karcutskie et al. JAMA Surg. 2017; Templin et al. N Engl J Med. 2015; Nor et al. Lancet Neurol. 2005]. However, we acknowledge that this is a simplified method widely used in biomedical research published in top medical journals to select the variables included in the multivariable analysis. We agree that this method has considerable limitations and exposes to the real risk of missing important elements as there are variables that may not be important in a univariable association but are important in the full model. [Sun et al. J Clin Epidemiol 1996; Harrel et al. Regression Modeling Strategies: With Applications to Linear Models, Logistic Regression and Survival Analysis. New York, NY: Springer-Verlag; 2001; Heinze et al. Transpl Int 2017]

In fact, there are lots of reasons to include non-significant variables as they mediate and moderate other relationships, theory says that they ought to be in the model, the effect size is large, even if it is not significant, it was part of your a priori hypothesis, etc. However, the process of selecting variables is very arduous and there is no agreement on the strategy to be used, as the most appropriate way can vary greatly according to different problems and disciplines. Commonly used methods for Cox regression model are forward selection, backward elimination, stepwise selection, purposeful selection algorithms, best subset selections and least absolute shrinkage and selection operator. [Bursac et al. Source Code Biol Med. 2008; Garcia et al. Biometrics. 2010].

Some methodologists suggest including all clinical and other relevant variables in the model regardless of their significance in order to control for confounding. This approach, however, can lead to numerically unstable estimates and large standard errors. Nevertheless, for this article, after statistical review, as suggested, we decided to include in the multivariable analysis all variables analyzed in the univariable, regardless of their significance, as they are all potentially clinically relevant. As a result, we found radiotherapy as a significant variable on multivariable analysis, although not significant on univariable. This is not an uncommon occurrence in this type of analysis, as highlighted in the literature [Lo et al. Changgeng Yi Xue Za Zhi. 1995; Wang et al. Shanghai Arch Psychiatry. 2017] and can be seen as evidence of the power of multivariable analysis in detecting confounding factors.

We would like to conclude by thanking you again for this comment which has led to a considerable improvement in the quality of the evidence for this study.

The manuscript has been modified accordingly.

  • In the discussion, the authors postulate that radiological classification did not have an effect on overall survival due to the protective effect of chemotherapy. This is far-reached, the simple explanation is that vessel proximity has nothing to do with the biological aggressiveness of the tumour, which is the major factor behind patient survival. 

Thank you for this comment. This part has been modified within the discussion section.

Round 2

Reviewer 1 Report

  • Please use competing risk analyses for LR as one patient cannot have a LR when dead of disease and 40% is dead of disease after 5 years

Thank you very much for the interesting observation, which allows us to formulate our opinion on the interpretation of these data. In standard survival analysis, the event-free time of subjects who do not experience the outcome of interest during the observation period is generally censored at the end of follow-up. However, censoring may occur for various reasons. A patient may be lost to follow-up during the study or may experience another event that makes further follow-up impossible or unnecessary. An important assumption of standard survival analysis methods such as the Kaplan-Meier method is that censoring is "independent" [Lau B et al. Competing risk regression models for epidemiologic data, Am J Epidemiol, 2009] of the event of interest. Based on this assumption, we censored patients who died during follow-up, patients who were lost to followup, and patients who underwent amputations for reasons unrelated to recurrence, considering all these reasons to be unrelated to local recurrence. In fact, in our opinion, local recurrence may be associated with overall survival as a predictive factor, not as an independent agent of death. Therefore, we believe that a competing cumulative incidence risk method is not necessary to provide a reliable estimate of the risk of recurrence in the context of univariate unadjusted analyses of prognostic factors. Nevertheless, we agree that the description of the analysis of the estimated local recurrence risk in the methods section appears to be inaccurate in its current form. Therefore, we modified it within the methods section.

Thanks for the kind reply on methodology. I believe we would need to agree to disagree on this as I firmly believe that survival analyses are in need for competing risk to control for survival bias: To predict the unadjusted probability of a certain outcome to occur, one can use the Kaplan–Meier method. However, in the presence of competing risks, using the Kaplan–Meier method is problematic. The method can handle only one single event at a time: all other events are treated as censored observations and the complement of the Kaplan–Meier estimate (1−KM) is interpreted as the probability of the event of interest in a hypothetical world in which the competing event does not exist. This kind of interpretation is not realistic in clinical practice. The independent censoring assumption is violated, meaning that the patients who experience a competing event at a given time often do not have the same chance of developing the event of interest after that time as the patients who are continued to be followed-up. As a result, the Kaplan–Meier method generally overestimates the probability of the event of interest and thus yields misleading results in the presence of competing risks.

  • Please use R0-1-2 margin descriptions as they are probably more informative than Ennekings and more predictive for LR and OS Margins were re-classified using the Residual tumour classification, according to this comment.
  • Please inform the reader on the multivariate analyses in what parameters are in the analyses as now it seems as if RxTh and margins are not in the Multivariate analyses. Thank you for this comment. The methods section has been modified accordingly.
  • Was the use of chemotherapy a predictive value due to inclusion bias ? it seems that healthy and young patients would indeed get Chemo and the elderly that are more likely to die within 5 years are not ? Chemotherapy was used mostly in non-infiltrative histotypes, in patients with metastasis at presentation, in grade 3 tumors, and in those presenting with a type 3-4 vessels involvement. These selection biases may explain why patients who underwent ChT had worse survival. This point has been discussed within the manuscript.

The influence of age and patient general health status needs to be in the discussion as patients over 65 generally are not offered ChTh / RxTh https://repub.eur.nl/pub/132372/Repub_133561_O-A.pdf

  • Please elaborate on the cross correlation between size and type and relation to the vessels? Thank you for this comment. This part was added to the results.
  • One could also argue that high-grade, infiltrative large tumors have an intrinsic higher chance of being close to the vessels and therefore need either resection of the vessels or (neo)adjuvant RxTh/Chemo to improve the outcome both on Local control and OS? And maybe it’s not the relation between vessels and STS but the biological intrinsic risk of the 2 STS that is displayed on MRI by being close to the vessels tumour, which is the major factor behind patient survival.

Thank you for this comment. Large and infiltrative STS were found to be associated with closeness to the major vessels. However, no correlation was found regarding FNLCC grade. As highlighted by multivariate analysis, we observed that proximity to major vessels can predict the risk of LR but not OS.

But size is not in the Multivariate analyses ? So we may just be looking at the size effect?

On the other hand, biological aggressiveness rather than closeness to vessels can predict survival. This part has been discussed within the manuscript.

But ageis not in the Multivariate analyses ? So we may just be looking at the age effect?

  • I would love to see a clear analysis of the influence of RxTh and vessel reconstruction on the OS and LR rates in order to present well-balanced advice for the readers on when to resect and when just to add Rxth to get better Local control when the tumor is close to vessels.

Thank you for this comment which has led us to a deeper reflection on the statistical analysis conducted. Initially, we did not include radiotherapy in the analysis because of its non-significance in the univariable analysis [Karcutskie et al. JAMA Surg. 2017; Templin et al. N Engl J Med. 2015; Nor et al. Lancet Neurol. 2005]. However, we acknowledge that this is a simplified method widely used in biomedical research published in top medical journals to select the variables included in the multivariable analysis. We agree that this method has considerable limitations and exposes to the real risk of missing important elements as there are variables that may not be important in a univariable association but are important in the full model. [Sun et al. J Clin Epidemiol 1996; Harrel et al. Regression Modeling Strategies: With Applications to Linear Models, Logistic Regression and Survival Analysis. New York, NY: Springer-Verlag; 2001; Heinze et al. Transpl Int 2017] In fact, there are lots of reasons to include non-significant variables as they mediate and moderate other relationships, theory says that they ought to be in the model, the effect size is large, even if it is not significant, it was part of your a priori hypothesis, etc. However, the process of selecting variables is very arduous and there is no agreement on the strategy to be used, as the most appropriate way can vary greatly according to different problems and disciplines. Commonly used methods for Cox regression model are forward selection, backward elimination, stepwise selection, purposeful selection algorithms, best subset selections and least absolute shrinkage and selection operator. [Bursac et al. Source Code Biol Med. 2008; Garcia et al. Biometrics. 2010]. Some methodologists suggest including all clinical and other relevant variables in the model regardless of their significance in order to control for confounding. This approach, however, can lead to numerically unstable estimates and large standard errors. Nevertheless, for this article, after statistical review, as suggested, we decided to include in the multivariable analysis all variables analyzed in the univariable, regardless of their significance, as they are all potentially clinically relevant. As a result, we found radiotherapy as a significant variable on multivariable analysis, although not significant on univariable. This is not an uncommon occurrence in this type of analysis, as highlighted in the literature [Lo et al. Changgeng Yi Xue Za Zhi. 1995; Wang et al. Shanghai Arch Psychiatry. 2017] and can be seen as evidence of the power of multivariable analysis in detecting confounding factors.

But no comments are made on the effect of size and rxth in the advise on the treatment of STS in the presence of vessel proximity…

We would like to conclude by thanking you again for this comment which has led to a considerable improvement in the quality of the evidence for this study. 3 The manuscript has been modified accordingly.

  • Also close to the vein is different from close to artery ?? Thank you again for the precious comment. In this retrospective analysis, artery and vein involvement were considered together as “major vessels involvement”. In the case of vascular bypass, all cases had only artery reconstructed. Thus, we believe that the identification of artery involvement by STS is more important.

Author Response

We thank the reviewer for the precious comments.

Multivariate analysis was further developed, including age of the patient and size of the tumor in the analysis.

The influence of age on survival has been discussed with a related reference.

The effects of RTE on local recurrence was also discussed.

Reviewer 2 Report

In this revised version authors have done a considerable effort and improved the paper. The conclusions are now clearly supported by the results, and the findings are useful in the clinical setting. 

Author Response

We thank the reviewer for the appreciation of our efforts to ameliorate the paper.